# *Escherichia coli* Increases its ATP Concentration in Weakly Acidic Environments Principally through the Glycolytic Pathway

**DOI:** 10.3390/genes11090991

**Published:** 2020-08-25

**Authors:** Wenbin Zhang, Xin Chen, Wei Sun, Tao Nie, Natalie Quanquin, Yirong Sun

**Affiliations:** 1Guangdong Provincial Key Laboratory of Protein Function and Regulation in Agricultural Organisms, College of Life Sciences, South China Agricultural University, Guangzhou 510640, China; zhangwenbin@scau.edu.cn; 2Guangzhou Institutes of Biomedicine and Health, Chinese Academy of Sciences, Guangzhou 510530, China; weis2016@126.com (W.S.); nietaoly@126.com (T.N.); 3Guangdong Key Laboratory of IoT Information Technology, School of Automation, Guangdong University of Technology, Guangzhou 510006, China; xinchen@gdut.edu.cn; 4Department of Microbiology, Immunology and Molecular Genetics, University of California, Los Angeles, CA 90095, USA; vashkoda@gmail.com

**Keywords:** *Escherichia coli*, acid resistance, ATP, glycolysis, energy metabolism

## Abstract

Acid resistance is an intrinsic characteristic of intestinal bacteria in order to survive passage through the stomach. Adenosine triphosphate (ATP), the ubiquitous chemical used to power metabolic reactions, activate signaling cascades, and form precursors of nucleic acids, was also found to be associated with the survival of *Escherichia coli* (*E. coli*) in acidic environments. The metabolic pathway responsible for elevating the level of ATP inside these bacteria during acid adaptation has been unclear. *E. coli* uses several mechanisms of ATP production, including oxidative phosphorylation, glycolysis and the oxidation of organic compounds. To uncover which is primarily used during adaptation to acidic conditions, we broadly analyzed the levels of gene transcription of multiple *E. coli* metabolic pathway components. Our findings confirmed that the primary producers of ATP in *E. coli* undergoing mild acidic stress are the glycolytic enzymes Glk, PykF and Pgk, which are also essential for survival under markedly acidic conditions. By contrast, the transcription of genes related to oxidative phosphorylation was downregulated, despite it being the major producer of ATP in neutral pH environments.

## 1. Introduction

Both commensal and pathogenic enteric bacteria have adapted to resist several hours of exposure to the acidic environment of the human stomach, which has an average pH of 2.0, before entering the more hospitable intestinal tract. Multiple genes and pathways play a role in acid resistance (AR) and the acid tolerance response (ATR) in bacteria [1]. At least four AR systems have been identified in *Escherichia coli* [2,3,4,5,6], which make use of enzymatic processes to consume protons or produce metabolites to buffer the pH and protect against stress [7,8]. AR1 (oxidative system) requires the sigma factor RpoS [9,10] and the cyclic AMP receptor protein CRP [11]. AR2–AR4 function through amino acid decarboxylation [4,12,13], with AR2, AR3 and AR4 reliant on glutamate (glutamate decarboxylase system) [12,13], arginine (arginine decarboxylase system) and lysine (lysine decarboxylase system) [14,15,16,17], respectively. AR2 is the most effective amino acid-dependent system, requiring two glutamate decarboxylases (GadA and GadB), which are active at an acidic pH, and the antiporter GadC [18,19,20]. The glutaminase and adenosine deaminases were also shown to contribute to AR in *E. coli*, and these enzymes use their respective substrates to release NH_3_ into the cytoplasm and raise the intracellular pH [21,22].

Interestingly, we previously published findings demonstrating a relationship in *E. coli* between intracellular ATP levels and the environmental pH [23]. We found that a high concentration of ATP within the bacterial cell is necessary for survival under extremely acidic conditions [23]. Although ATP levels decreased rapidly when the media was adjusted from a near-physiological pH of 7.5 to 2.5, ATP concentrations were actually found to increase only when cells were pre-adapted to a weakly acidic environment (pH 5.5), which was still permissible to *E. coli* growth [23].

The mechanism and purpose of this observed effect remain unclear, although there have been several proposed explanations. One possibility is that under stress at low pH, metabolic processes that would otherwise consume ATP are intentionally suppressed or forcibly inhibited by suboptimal enzymatic conditions. Bacterial growth was in fact noted to be reduced at pH 5.5. Alternatively, the production of ATP in weakly acidic environments may be augmented. ATP is produced in *E. coli* through both oxidative phosphorylation and glycolysis when glucose is present as the carbon source.

In oxidative phosphorylation, F_1_Fo-ATPase catalyzes the synthesis of ATP from ADP and inorganic phosphate using the electro-chemical gradient of protons across the cellular membrane. The greater pH gradient between the intracellular and extracellular compartments at pH 5.5 could boost the membrane potential and drive further ATP synthesis. Interestingly, when compared to levels at pH 7.5, both the pH gradient and membrane potential between intracellular and extracellular compartments were found to be minimal at a low pH (pH 2–2.5) [2,3,24]. Our previous work had shown that ATPase mutants could still elevate the level of ATP at pH 5.5 [25]. Foster et al. speculated that the ATPase may function as a proton pump and consume ATP to regulate the intracellular pH in *E. coli* [7]. During glycolysis, one mole of glucose is fermented and at least two moles of ATP are synthesized, which can either be used for biosynthesis or be hydrolyzed by F_1_Fo-ATPase [26]. These results suggest the possibility that the ATPase consumes ATP to pump out protons in acidic environments, and that another mechanism independent from the ATPase is used to generate ATP.

In glycolysis, glucose is converted to pyruvate through a 10-step pathway with multiple intermediate metabolites (Figure 1). Steps 1 and 3 consume ATP, and steps 7 and 10 catalyze ATP synthesis, resulting in a net balance of 2 ATP per molecule of glucose. The initial step of the glycolysis pathway is catalyzed by glucokinase (Glk), a hexokinase isozyme that facilitates phosphorylation of glucose to glucose-6-phosphate. Step 7 is catalyzed by 3-phosphoglycerate kinase (Pgk) [27,28]. Two isozymes of pyruvate kinase, PykF and PykA, oversee the final step, where phosphoenolpyruvate is converted into pyruvate [29,30,31,32]. Pyruvate is then used in the tricarboxylic acid (TCA) cycle under aerobic conditions to produce additional NADH and succinate for further processing in oxidative phosphorylation. Citrate synthase (GltA) catalyzes an early step in the TCA cycle to condense oxaloacetate and acetyl coenzyme A into citrate and coenzyme A [33,34].

In this study, we attempt to clarify which metabolic pathway or genes encoding enzymes are responsible for the increase in intracellular ATP concentration in *E. coli* grown in weakly acidic media with glucose as the sole carbon source. This will in turn reveal the mechanisms used by *E. coli* for adaptation to acidic environments. The expression of genes encoding enzymes responsible for ATP synthesis at different pH levels was analyzed, and the effects on AR and ATP concentration were investigated.

## 2. Materials and Methods

### 2.1. Strains, Plasmids, Culture Media and Reagents

The bacterial strains and plasmids used in this study are listed in Table 1. All strains were initially grown at 37 °C in LB medium, then tested in EG medium, which is E medium [35] containing 0.4% glucose [36]. The medium pH was adjusted by the addition of NaOH or HCl. Antibiotics were used at the following concentrations: ampicillin, 100 µg/mL; kanamycin, 30 µg/mL; chloramphenicol, 30 μg/mL. Antibiotics, L-arabinose, *N*,*N*’-dicyclohexylcarbodiimide (DCCD) and carbonyl cyanide 3-chlorophenylhydrazone (CCCP) were obtained from Sigma-Aldrich (St. Louis, MO, USA).

### 2.2. One-Step Inactivation of Chromosomal Genes in E. coli

Gene knock-out mutants in the glycolysis pathway in *E. coli* were created using conventional Red-mediated recombination [37]. A linear fragment containing the kanamycin or chloramphenicol resistance gene flanked by about 60 base pairs of the target metabolic gene was introduced through electroporation into *E. coli* strain BW25113, which already carried the Red helper plasmid pKD46. The primers used to amplify these genes are shown in Table 2. L-arabinose (1 mM) was added to the medium and bacteria were incubated at 37 °C for 1 h after electroporation. The mutants were selected on plates of LB medium containing the relevant antibiotic. DNA from individual transformants was isolated and tested by PCR amplification to confirm the integration of the resistance gene, and the Red helper plasmid pKD46 was cured by growth at 42 °C [38]. Primers used for confirmation are shown in Table 2.

### 2.3. P1 Transduction

The knock-out genes were transferred from BW25113 to W3110 by P1 phage as previously described [39]. The transductants of W3110 were selected by antibiotics.

### 2.4. Acid Tolerance Response (ATR) and Acid Resistance (AR) Test

For the logarithmic phase ATR test, the survival of wild-type and mutant strains was measured as previously described [23,25]. After overnight culture in LB medium with antibiotics where necessary, the cells were diluted 1000-fold with EG medium at pH 7.5 and cultured at 37 °C until the optical density at 600 nm (OD_600_) reached 0.3 to 0.4. Cells were collected by centrifugation at 5000× *g* for 4 min and pellets were suspended in a 2-fold volume of EG medium at pH 5.5 before incubation under micro-aerobic culture conditions for 4 h without shaking. The OD_600_ reached 0.2–0.3 after a 4 h adaptation at pH 5.5. The adapted cells were washed with fresh EG medium at pH 5.5 and then diluted 100-fold in EG medium at pH 2.5 [40]. After incubation at 37 °C for 1 h or 2 h, the cells were spun down, resuspended in phosphate-buffered saline (137 mM NaCl, 2.7 mM KCl, 4.3 mM Na_2_HPO_4_, and 1.4 mM KH_2_PO_4_, pH 7.4) and spread on LB agar plates for overnight culture. Colonies were counted, and viability was expressed as the percentage of viable cells out of the total cell number before the acidic challenge. The tests were performed in at least three independent experiments with biological triplicates.

For stationary phase AR assays, cells were grown overnight in LB buffered with either 100 mM morpholinepropanesulfonic acid (MOPS, pH 8) or 100 mM morpholineethanesulfonic acid (MES, pH 5.5) [10]. Cultures were grown in 4 mL of the buffered medium in 15-mm test tubes with shaking (220 rpm) at 37 °C to stationary phase (22 h) followed by 1:1000 dilution into prewarmed (37 °C) pH 2.5 EG. Viable-cell counts were determined at 2 h post-acid challenge by diluting cells in PBS, plating cells onto LB agar, and incubating plates at 37 °C before counting the number of colonies. Values given are representative of the results of triplicate experiments reproducible to within 50%.

### 2.5. ATP Content Measurement

After culturing as above, the cells were chilled on ice and then centrifuged at 10,000× *g* for 5 min at 4 °C. The pellets were treated with a solution containing 20 mM Tris-HCl, 50 mM MgSO_4_, 4 mM EDTA and 50% methanol at pH 7.4 for 30 min at 70 °C and then were centrifuged at 10,000× *g* for 5 min. The ATP content of the supernatant was measured as described previously by a luminometer (Turner Designs, Inc., Sunnyvale, CA, USA) [23,41,42]. Luciferase and standard ATP were purchased from Sigma.

The measurement was repeated at least three times independently, and the mean value and the standard deviation were calculated.

### 2.6. RNA Extraction and RNA-seq Analysis

The cells were cultured at pH 7.5 in EG medium until the OD_600_ reached 0.3–0.4, then spun at 10,000× *g* for 5 min and transferred into the same volume of fresh EG medium at either pH 7.5 or pH 5.5. After half hour incubation, total RNA was isolated using RNA-*Solv* reagent (Omega, Norcross, GA, USA) according to the manufacturer’s protocol. The concentration and purity of RNA were determined using the GeneQuant RNA/DNA Calculator (Pharmacia-Biotech, Cambridge, UK). The RNA samples were stored at −80 °C for the RNA-seq assay. The isolated RNA was sequenced on an Illumina HiSeq 2000 at Ribobio Co (Guangzhou, China). Reads contaminated by adapter sequence were removed from the pair end (PE) reads (30 bp) when either of the PE reads was polluted. Low quality reads (phred quality ≤19 in more than 15% of all the read bases) were further filtered out. Any reads that aligned to rRNA sequence were also filtered out. In addition, we removed reads that contained more than 5% Ns. The cleaned reads were aligned to the reference sequences in the National Center for Biotechnology Information (NCBI) database by Rockhopper v2.03 using default parameters. HTSeq v0.6.0 was utilized to calculate the co gen units of each e, and gene expression levels were represented by the Reads per Kilobase per Million reads (RPKM) method, as described by Mortazavi et al. [43]. The RNA-seq analyses were repeated in two independent experiments using three biological replicates. Differentially expressed genes (DEGs) at either pH 5.5 or pH 7.5 were identified using the DEGseq method (software: DESeq2), using a cut off of fold change > 2 and *p* value < 0.01.

### 2.7. Gene Ontology Annotation and KEGG Pathway Analysis

Gene Ontology (GO) is comprised of three aspects to describe gene functions: biological process (BP), molecular function (MF) and cellular component (CC). When performing functional enrichment analysis on the DEGs, we considered the BP branch. The Kyoto Encyclopedia of Genes and Genomes (KEGG) PATHWAY database was referenced for their maps of molecular interactions and reaction and relation networks for metabolism.

We used the online web tool DAVID [44] for functional enrichment analysis of GO using the KEGG pathways. EASE score was used to evaluate whether the DEGs were significantly enriched for a specific gene function. Benjamini–Hochberg (BH) method was used to adjust *p*-values for multiple comparisons. The R software programs fisher.test and p.adjust was also used. The enrichment threshold for *p*-value significance was set to 0.01.

### 2.8. Other Methods

The growth curves of all strains were measured by Bioscreen C (Oy Growth Curves Ab Ltd. Helsinki, Finland). Complementation plasmids were cloned as described previously [23]. After overnight culture in LB medium with antibiotics where necessary, the cells were diluted 500-fold with EG medium at pH 7.5 or pH 5.5 and cultured at 37 °C. Protein concentrations were measured by absorption at 595 nm using the Bradford Protein Assay (BioRad, Bradford, UK), with bovine serum albumin as the standard. The mRNA level was measured by qPCR following methods described in previous publications [21]. Primers for qPCR are listed in Table 2.

### 2.9. Statistics

Data were reported as mean ± standard error of the mean. Statistical significance of difference was determined by using unpaired two-tailed Student’s *t*-test. A value of *p* < 0.05 was considered to be statistically significant. One-way ANOVA(Analysis of Variance) followed by Bonferroni’s post hoc test or F-test was used to determine significant differences (*p* < 0.05) between groups.

## 3. Results

### 3.1. Carbonyl Cyanide 3-Chlorophenylhydrazone (CCCP) and *N*,*N*′-Dicyclohexylcarbodiimide (DCCD) Affect the ATP Level and Acid Resistance under Different Conditions

We had previously reported that *E. coli* mutants DK8 (deletions in all subunit genes of the FoF1-ATPase), *atpE* (deleted subunit c of the FoF1-ATPase) and *atpD* (deleted subunit β of the FoF1-ATPase) did not show significant changes in intracellular ATP concentrations under weakly acidic conditions [25]. CCCP is a protonophore and an uncoupler of oxidative phosphorylation that disrupts the cell membrane potential [45]. To test the survival of *E. coli* under different acidic conditions when ATP production via oxidative phosphorylation is inhibited, *E. coli* strain W3110 was grown in EG medium at pH 7.5 to an OD_600_ of 0.3 to 0.4. The cells were then steadily adapted to increasing acidic conditions by growing at pH 5.5 for 4 h and then challenging for 1 h at pH 2.5, with the addition of CCCP (30 μM) at different time points (Figure 2A). No viable cells were detected when W3110 was given CCCP during the exposure to a pH of 2.5, whether or not CCCP had also been given at pH 5.5. However, there was no significant effect on survival when CCCP was only added to cells at a pH of 5.5 (Figure 2A). Interestingly, the ATP level was only significantly decreased after the addition of CCCP at pH 7.5, with no significant decreases noted at pH 5.5 (Figure 2B).

DCCD, an inhibitor of F_0_F_1_-ATP synthase, binds to the H^+^-transporting acidic residue of the F_0_ c subunit, preventing the ATP synthesis activity of F_1_ [46]. We found that the survival of *E. coli* under extremely acidic conditions (pH 2.5) was decreased by the addition of increasing amounts of DCCD (Figure 2C). Interestingly, intracellular ATP levels were decreased at pH 7.5, but not significantly affected by DCCD at pH 5.5, except at the highest concentration (0.5 mM) (Figure 2D). Recently, it was reported that DCCD inhibits F_0_F_1_-ATPase activity during the fermentation of glycerol through a process dependent on pH and potassium ions [26,47].

These results suggest that under weakly acidic conditions, ATP production via oxidative phosphorylation is either minimal or can easily be compensated through other processes.

### 3.2. Global Identification and Functional Inference of Acid-Regulated Genes, and Transcription Analysis of Known Glycolysis Genes

We analyzed the changes in the transcription levels of *E. coli* genes and transcripts, which include most of those with known roles in functional and metabolic pathways, both before and after the adaptation of cells to mildly acidic media [5,13,48]. Our RNA-seq data showed that the majority of gene transcription was upregulated (379 genes) rather than downregulated (261 genes) in *E. coli* undergoing acid adaptation at pH 5.5. A total of 155 genes were induced by at least two-fold (*p* < 0.01, FDR < 0.05), and interestingly, 26 of those genes were induced to almost four-fold higher levels than their baseline (Figure 3A, Appendix A). By contrast, the transcription of 69 genes was decreased by two-fold or more (*p* < 0.01, FDR < 0.05) (Figure 3A, Appendix A). Among the induced genes, several belonged to metabolic pathways known to be associated with the AR2, AR3 and AR4 systems (Figure 3B, Appendix A). Genes whose transcription was downregulated included ones associated with fatty acid oxidation, the electron transport chain, “energy derivation by oxidation of other organic compounds” and “generation of precursor metabolites producing energy”, indicating that under weakly acidic conditions, *E. coli* may not rely significantly upon these other systems for energy production (Figure 3C). This includes the *sdhC* gene, which encodes succinate dehydrogenase complex subunit C, a major component of the TCA cycle (Appendix A).

The expression of genes encoding glycolytic enzymes such as the acetyltransferase component of the pyruvate dehydrogenase complex (AceF) was upregulated greater than two-fold, and pyruvate kinase I (PykF) was upregulated almost two-fold (Appendix A). The expression of genes in the glycolysis pathway was further tested by qPCR. The results showed that under weakly acidic conditions, *pykF* and *glk* expression was increased 2- and 1.5-fold, respectively, compared to at a neutral pH (Figure 3D). Other genes in the glycolysis pathway (*pykA*, *pgi*, *pgk* and *pfo*), and the first TCA cycle gene (*gltA*), were also upregulated under weakly acidic conditions (Figure 3D).

Altogether, the gene transcription data suggest that the expression of several glycolysis pathway genes increases in weakly acidic environments, and that the increased ATP content is less likely due to the induction of other metabolic pathways or reduced energy consumption.

### 3.3. Characteristics of E. coli Knockout Mutants in Glycolysis and TCA Cycle Genes

As glycolysis gene transcription was upregulated under weakly acid conditions, we created knockout mutants of key enzymes in the glycolytic pathway whose reactions directly catalyze ATP synthesis (*pgk, pykA* and *pykF*), as well as other glycolysis genes (*pgi*, *glk* and *pfo*) and the enzyme responsible for initiating the TCA cycle (*gltA*) [49]. Using conventional Red-mediated recombination, these genes in *E. coli* strain BW25113 were replaced with chloramphenicol or kanamycin resistance genes, and clones were selected on plates with the appropriate antibiotic. Individual knockouts were confirmed through PCR amplification.

The growth kinetics of these mutants under different conditions was compared. The *ΔgltA* and *Δglk* strains grew slower than the other mutants in EG medium (Figure 4A) at pH 7.5. Although *ΔgltA* exhibited the slowest growth rate at a neutral pH in EG (Figure 4A,C) and LB media [49], *Δglk* also grew to a low optical density at 600nm (OD_600_) in EG media at both pH 7.5 and 5.5 (Figure 4A,B). At pH 5.5, The *glk* mutant reached an OD_600_ of only 0.2 with the lowest growth rate (μ) of all the mutants, likely because this key gene is responsible for the first step in the glucose metabolism pathway. The *ΔpykA* and *ΔpykF* mutants still grew at pH 7.5 and pH 5.5 in EG medium (Figure 4), but a double-knockout mutant of both genes did not (Figure 4). *ΔpykA* grew slower than *ΔpykF* at pH 7.5 (Figure 4A,C), however, this trend was reversed at pH 5.5 (Figure 4B,D). These data indicate that the two pyruvate kinases are necessary for *E. coli* growth in EG medium, in which glucose is the sole energy source, and that *pykF* and *pykA* are particularly active in acidic and neutral pH environments, respectively. In EG medium at pH 5.5, *Δpgk* reached steady-state slowly, with a final optical density of around 0.4, almost as low as that of *ΔpykF,* which also shared the same μ value (Figure 4B,D). These results suggest that GltA may be important for *E. coli* growth. Under mildly acidic conditions, components of the glycolysis pathway (Glk, Pgk and PykF) take on a more important role. Other mutants such as *Δpgi* and *Δpfo* showed no effect on the growth of *E*. *coli* compared to wild type strain. Our results indicate that the metabolism of glucose by *E. coli* in different pH environments may be mediated by different glycolytic enzymes.

### 3.4. Comparisons of ATP Levels between Different Mutant Strains under Weakly Acidic Conditions

It was noted that the ATP concentration in *E. coli* was increased after adaptation to weakly acidic conditions [23]. To investigate which glycolysis pathway genes are important for this process, we compared the ATP content in our knockout mutants and the wild-type strain W3110 at pH 7.5 and pH 5.5 in EG medium (Figure 5A). The increase in ATP after adaptation to pH 5.5 was slightly higher in *ΔpykA* compared to W3110. The *gltA* mutant showed a slightly higher ATP content after acid adaptation, however, it did not reach the level seen in W3110.

In contrast to the effect seen in W3110, adaptation to pH 5.5 resulted in a drop in ATP level for deletion mutants *Δglk*, *ΔpykF* and *Δpgk*. The *Δglk* mutant also showed significantly lower ATP content compared to W3110 at pH 7.5. Complementation using plasmids harboring the *pykF* and *pgk* genes resulted in wild-type ATP levels in their respective deletion mutants (Figure 6A).

These results indicate that the glycolysis pathway is the main source of ATP production in *E. coli* in acidic environments. In addition, the enzymes encoded by *pgk* and *pykF* may be important for increasing ATP synthesis under weakly acidic conditions. It is possible that an alternative metabolic system produces ATP in *pgk* and *pykF* mutants, but it may be less active or fail to increase ATP under acidic conditions.

### 3.5. Comparisons of Acid Resistance between Different Mutant Strains

As ATP content is important for *E. coli* survival in extremely acidic environments, these mutants were also tested for acid resistance. Survival after the acid challenge was reduced in all strains, but compared to the wild-type, the effect was particularly significant in *glk*, *pgk* and *pykF* mutants, with a smaller difference seen in the *gltA* mutant (Figure 5B). Complementation in mutants, *Δpgk* and *ΔpykF* with their respective plasmids restored the level of AR to that of the wild-type strain (Figure 6B). Together with our previous results, these data show that the glycolysis genes *glk*, *pgk* and *pykF* participate in regulating ATP levels in *E. coli* to enable their survival upon acid challenge.

We next examined whether these genes also affect acid resistance for cells in the stationary phase of growth. When the mutants were grown to stationary phase at pH 8 and then challenged for 2 h at pH 2.5, only *ΔpykF* showed significantly decreased survival (*p* < 0.05) (Figure 5C). When the mutants were grown to stationary phase at pH 5.5 before challenging for 2 h at pH 2.5, both the *pykF* and *pgk* mutants showed lower survival. We noted that wild-type cells normally had a higher ATP content at pH 5.5 than that at pH 8 at the stationary phase. However, the intracellular ATP level of *ΔpykF* was significantly decreased in the stationary phase at both pH 8 and pH 5.5 (Figure 5D), while the *pgk* mutant showed a significantly decreased level only when grown to stationary phase at pH 5.5. These results confirmed that the glycolysis pathway mainly functions to supply ATP at a weakly acidic pH.

To further investigate whether these mutants primarily affect AR and ATP levels through the glycolysis pathway, fructose and pyruvate were used to bypass the *glk* and *pykF* mutations. The data showed that 0.2% fructose added to 0.2% glucose could restore ATP levels and cell survival of *glk* mutants under extremely acidic conditions (Figure 6C,D). However, 0.4% pyruvate added to 0.2% glucose as a carbon source was unable to rescue *pykF* mutants, and even wild-type W3110 showed lower ATP levels and lower survival (Figure 6D). These results confirm that genes in the glycolysis pathway, particularly those with a role in ATP generation, are responsible for keeping ATP levels elevated in weakly acidic environments.

## 4. Discussion

Gut bacteria primarily enter the human intestinal tract through the stomach, whose acidic environment might initially be buffered to a pH of up to 6.0 depending what other contents are being digested. After approximately 4 h, the pH gradually decreases to 2.5, granting the bacteria time to activate their acid resistance genes and adapt [50]. During this period of adaptation to weakly acidic conditions, *E. coli* was found to have an elevated ATP level, which is believed to play an important role in AR [23].

The concentration of ATP in *E. coli* cells is known to increase in response to certain stresses, such as the change in osmotic pressure and environmental pH [23,51]. The *pykA* gene was reported to negatively regulate ATP levels under anaerobic conditions [32]. The availability of ATP during adaptation to acidic conditions becomes increasingly more important for survival as the pH decreases [23]. This could be explained by an increased demand for energy as new mechanisms are activated to counteract the hostile environment and keep the cytoplasmic space at a constant pH [52,53]. However, these systems no longer fully compensate after the pH drops below 6, as the intracellular compartment will begin to acidify at that point [53]. For this reason, we selected a pH of 5.5 as our experimental condition for growth under a mildly acidic challenge [54].

Whether the increase in ATP levels during acid adaptation is a direct result of increased production (and if so, by which pathway), or through limiting the energy consumption of nonvital systems, was previously unclear. Our analysis showed that gene transcription was upregulated rather than downregulated in *E. coli* undergoing acid adaptation, suggesting that the mechanism is not via a reduction in ATP use. Furthermore, genes related to transcription were more upregulated than those related to translation under these conditions. The ATP content also correlated directly with the growth of different mutants and the wild type strain (Figure 4 and Figure 5).

Our results show that the glycolytic enzymes encoded by *pykF* and *pgk* provide the primary supply of ATP to *E. coli* under weakly acidic conditions. In addition, the gene expression level of AceF, a component of the pyruvate dehydrogenase complex, was also increased more than two-fold during acid adaptation. Similarly, the levels of enzymes involved in glucose metabolism pathways, including glucokinase (Glk), as well as the expression of pyruvate ferredoxin oxidoreductase (Pfo), were increased under acidic conditions. The deletion of *glk* showed significant effects on both AR and the ATP level, likely because Glk mediates the initial step of the glycolysis pathway and is a key enzyme. Its deletion may also affect the downstream ATP production from PykF and Pgk. By contrast, when we treated *E. coli* with CCCP to disrupt the membrane potential, the ATP level decreased only slightly during acid adaptation, revealing that oxidative phosphorylation is not a major supplier of ATP under these conditions, despite the fact that at near-neutral pH values, it is the main producer of ATP. *E. coli* treated with DCCD also showed no significant change in ATP level at pH 5.5, which is in agreement with our ATPase mutant data [25]. This was also supported by our RNA-seq data showing that the transcription of genes associated with respiration was downregulated in weakly acidic growth conditions. The transcription of genes in the fatty acid and lipid oxidation pathways, as well as those related to energy derivation by oxidation of other organic compounds or the generation of precursor metabolites, were also downregulated during adaptation to weakly acidic media (Figure 3C).

Our data suggest that glycolysis is the main pathway that supplies ATP at pH 5.5 and the metabolism of glucose by *E. coli* in different pH environments may be mediated by different glycolytic enzymes. *ΔpykA* grew slower than *ΔpykF* at pH 7.5 (Figure 4A), but, this trend was reversed at pH 5.5 (Figure 4B). *Δglk* grew at the slowest rate in EG media at both pH 7.5 and 5.5, yet it may be supported by other glucose metabolism pathways. For example, Roseman et al. reported that when glucose is taken up by *E. coli*, it can be converted to glucose 6-phosphate by phosphotransferase [55]. However, this enzyme activity may be inhibited by a low pH.

Our experiments also showed that the addition of pyruvate, the end-product of glycolysis, could not rescue *pykF* mutants or even wild-type W3110 at pH 5.5 even when 0.2% glucose was present as a carbon source (Figure 6C,D). In fact, it may be that pyruvate itself negatively impacts *E. coli* survival under extremely acidic conditions. Pyruvate is normally further metabolized to lactic acid, acetic acid, alanine, or acetyl-CoA through the TCA cycle [56,57]. We showed that adaptation to mildly acidic media (pH 5.5) resulted in a two-fold upregulated expression of the *aceF* pyruvate dehydrogenase gene and the lactate dehydrogenase *ldhA* gene (Appendix A). Acetyl-CoA is converted into acetate, coenzyme A, and ATP via the enzymes phosphotransacetylase (Pta) and acetate kinase (AckA). One study showed that the deletion of *ackA* and *pta* increased the survival of *E. coli* [58]. Others have found that when excess glucose is converted to acetate (as also happens with the addition of pyruvate or during phosphate starvation), it can partially uncouple and deplete the proton motive force (PMF), causing cell death under extremely acidic conditions [59,60,61]. In *E. coli*, The ATPase may pump out protons to regulate the intracellular pH [2], which could also compensate for the decrease in the PMF otherwise mediated by the respiratory chain. In our own experiments, we also observed that the addition of acetate decreased the survival of *E. coli* in extremely acidic conditions. It remains unclear whether this negative impact on survival is related to decreased efficiency in ATP production or to its direct effects on the PMF. GadY, which encodes a small RNA, has been reported to decrease acetate production [62] and our results showed that GadY was upregulated under weakly acidic conditions (Appendix A). These data could suggest that pyruvate metabolism under acidic conditions may be through the lactate pathway rather than the TCA cycle or acetate pathway.

It is generally accepted that amino acid decarboxylation can regulate intracellular pH in acidic environments [1,18]. Cells containing more ATP showed a higher intracellular pH (pHi) and greater cell survival in pH 2.5 medium than cells with a low ATP concentration [23]. Amino acids could help maintain the intracellular ATP level and increase cell survival [23]. Proton consumption by hydrogenase-3 in *E. coli* has been implicated in the ability to survive under extremely acidic and anaerobic conditions [63]. ATP may have multiple effects on acid resistance, with the different ATPases hydrolyzing it and regulating the pHi under different conditions [2,25,26]. We previously reported on a DNA repair system which uses ATP as the substrate, and is necessary for *E. coli* survival at a very low pH [23]. While the mechanism by which ATP helps *E. coli* survive extremely acidic environments remains unknown, the importance of maintaining an elevated level while growing under such conditions is clear.

Our analysis revealed that other metabolic processes can also be affected by acidic environments. Apt was highly upregulated, suggesting that adenosine nucleotides may be synthesized more actively under these conditions. Interestingly, this is in line with our previous study showing that PurA and PurB are important for AR [23]. The expression of genes associated with biofilms (MinD) and quorom sensing (LuxS) was also increased (Appendix A) [64,65], indicating that these processes may also be affected by environmental pH. The expression level of *gadX* [66] was increased at a weakly acidic pH, which was regulated by RpoS (Appendix A).

## 5. Conclusions

In this study, we demonstrated that *glk*, *pykF* and *pgk* were necessary for the rise in ATP under weakly acidic conditions and for survival in markedly acidic environments (typically pH 1.5–3.5). In contrast, the metabolic pathways related to oxidative phosphorylation, fatty acid oxidation and energy derivation by oxidation of other organic compounds were downregulated. The inhibition of oxidative phosphorylation did not affect the ATP increase at pH 5.5. These results showed that glycolysis is the primary source of the elevated ATP levels seen in *E. coli* grown under weakly acidic conditions.

## Figures and Tables

**Figure 1 genes-11-00991-f001:**
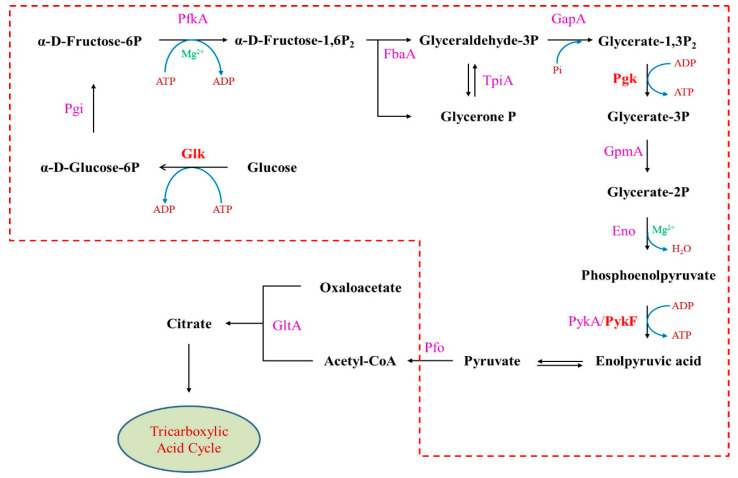
Outline of glycolysis and the initiation of the tricarboxylic acid (TCA) cycle. The 10 steps of glycolysis (outlined in red): step 1, hexokinase (Glk); step 2, glucose-6-phosphate isomerase (Pgi); step 3, phosphofructokinase-1 (PfkA); step 4, fructose 6-phosphate, fructose-bisphosphate aldolase (FbaA); triose-phosphate isomerase (TpiA); step 5, glyceraldehyde 3-phosphate dehydrogenase (GapA); step 6, phosphoglycerate kinase (Pgk); step 7, phosphoglycerate mutase (GpmA); step 8, phosphopyruvate hydratase (Eno); step 9, pyruvate kinase (PykA/PykF); step 10, pyruvate dehydrogenase (Pfo). Citrate synthase (GltA) catalyzes the initial step in the TCA cycle. P: phosphate; BP: bisphosphate.

**Figure 2 genes-11-00991-f002:**
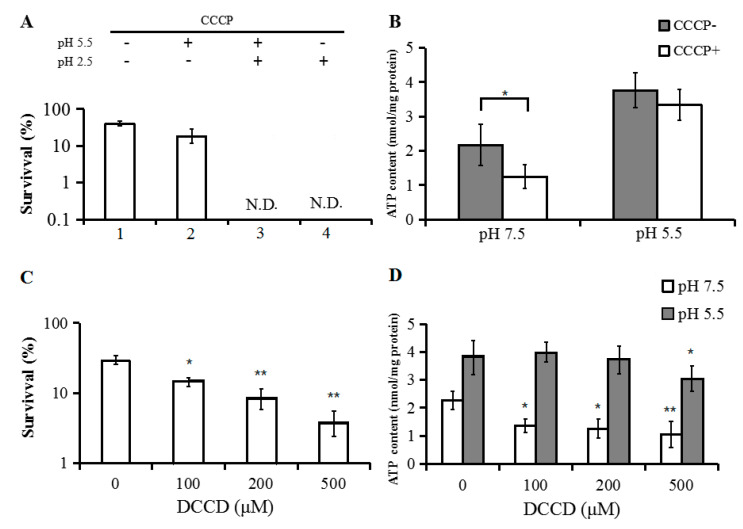
The effect of carbonyl cyanide 3-chlorophenylhydrazone (CCCP) and *N*,*N*′-Dicyclohexylcarbodiimide (DCCD) on cells survival and ATP levels in *E. coli*. (**A**) W3110 cells were adapted for 4 h at pH 5.5 before challenging for 1 h at pH 2.5. CCCP was added at adaptation (pH 5.5) and challenge (pH 2.5) as indicated (N.D., not detected). (1) Control: W3110 without CCCP at both pH 2.5 and pH 5.5; (2) CCCP was added at pH 5.5, then cells were collected, washed twice and transferred to pH 2.5 for challenge; (3) CCCP was added both at pH 5.5 and pH 2.5; (4) CCCP was added only at pH 2.5. (**B**) The ATP content was measured after W3110 cells were incubated at pH 5.5 and pH 7.5 for 4 h with and without CCCP (viable cell number~5 × 10^7^/Ml). (**C**) W3110 cells were adapted for 4 h at pH 5.5 before challenging for 1 h at pH 2.5. DCCD was added at different concentrations as indicated. (**D**) The ATP content was measured after W3110 cells were incubated at pH 5.5 and pH 7.5 for 4 h with and without DCCD. The average values and standard deviations were obtained from three experiments. Comparisons are made between samples with and without CCCP or DCCD for both survival and ATP content after 1h of challenge. (** *p* < 0.01; * *p* < 0.05).

**Figure 3 genes-11-00991-f003:**
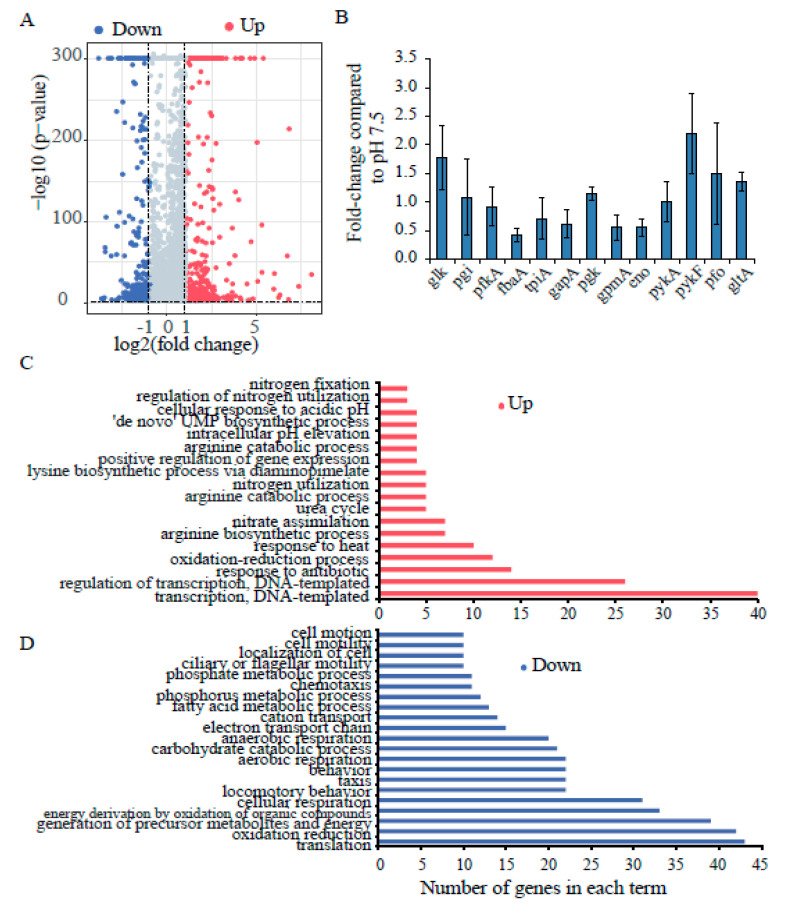
The transcription profile and Gene Ontology (GO) enrichment analysis of differentially expressed genes and gene expression in the glycolysis pathway. (**A**) Volcano plot representing the significance of the changes in gene expression from adaptation at pH 5.5 to 7.5. The red points represent upregulated genes, while blue points represent downregulated genes (fold change > 2, *p* < 0.01). The grey dots are the genes not differentially expressed. (**B**) Gene expression in the glycolysis pathway. The mRNA level of genes was measured by real-time PCR. The relative amount of gene mRNA to 16S rRNA was calculated. The values represent means ± standard deviations of the relative amounts obtained from three independent experiments. (**C**,**D**) The numbers of differentially transcribed genes for each GO term (biological process) are plotted. (**C**) GO terms enriched for upregulated genes. (**D**) GO terms enriched for downregulated genes. GO statistics were computed using the DAVID database.

**Figure 4 genes-11-00991-f004:**
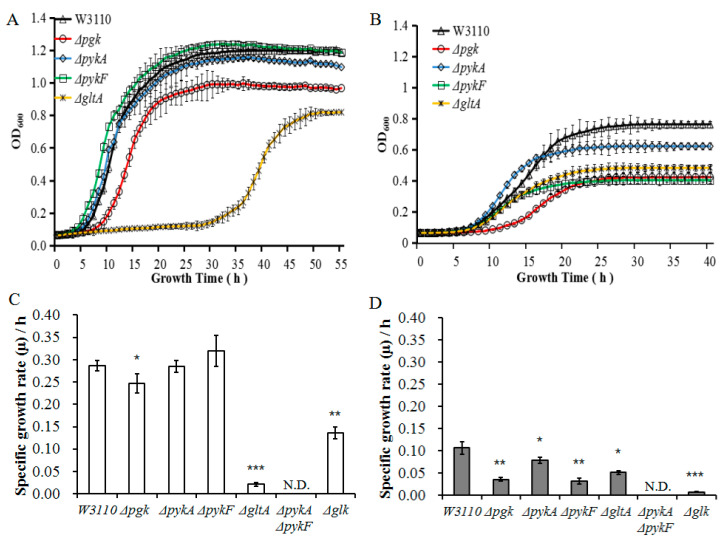
Growth characteristics of wild-type W3110 and gene knockout mutants. (**A**) Growth curves of different *E. coli* strains in EG medium at pH 7.5. The optical densities at 600 nm (OD_600_) were measured. (**B**) Growth curves of different *E. coli* strains in EG medium at pH 5.5. OD_600_ values represent the average of three separate experiments. (**C**) Specific growth rate (µ) of *E. coli* wild-type and mutants at pH 7.5. (**D**) Specific growth rate (µ) of *E. coli* wild-type and mutants at pH 5.5. N.D.: not detected. All OD600 and µ values represent the average of three separate experiments. Comparisons are made between the individual mutants and WT(wild type), asterisk (*) indicates *p*-value < 0.05, two asterisk (**) indicates *p*-value < 0.01, three asterisk (***) indicates *p*-value < 0.001.

**Figure 5 genes-11-00991-f005:**
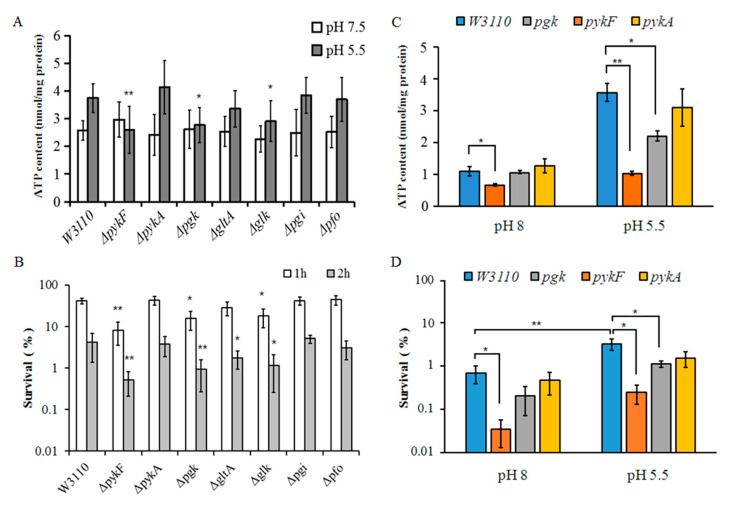
ATP concentration and acid resistance in wild-type W3110 and gene knockout mutants. (**A**) The concentration of ATP isolated from different *E. coli* strains at different pH values in EG medium. (**B**) Cells survival in different *E. coli* mutants and WT strain W3110. The average values and standard deviations obtained from three separate experiments are represented. (**C**) The concentration of ATP isolated from different *E. coli* strains at different pH values at stationary phase. (**D**) Acid resistance in different *E. coli* mutants and WT strain W3110 at stationary phase. Comparisons are made between the individual mutants and WT for both the survival and ATP content after 1h or 2hr challenge. (** *p* < 0.01; * *p* < 0.05).

**Figure 6 genes-11-00991-f006:**
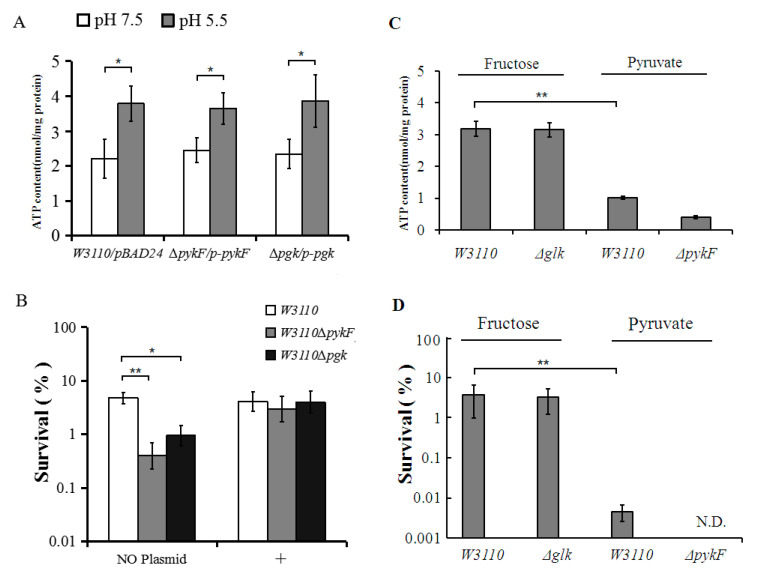
ATP level and cells survival after complementing or bypassing *pgk* and *pyk* knock-out mutations. (**A**) ATP levels after complementation of *pgk* and *pyk* knock-out mutants. (**B**) AR of *pgk* and *pyk* knock-out mutations complemented by plasmids (‘+‘) or not (“no plasmid”). (**C**) ATP levels at pH 5.5 after bypassing the *glk* and *pykF* mutations by using either fructose or pyruvate as the half of carbon source (0.2% glucose and 0.2% fructose or 0.4% pyruvate). (**D**) AR of cells bypassing the *glk* and *pykF* mutations. + Plasmid: W3110 harboring the pBAD24 vector; W3110*∆pykF* harboring pBAD24 with the *pykF* gene, and W3110*∆pgk* harboring pBAD24 with the *pgk* gene. Fructose: Fructose was added in E medium at a concentration of 0.2% with 0.2% glucose as the carbon source; Pyruvate: Pyruvate was added in E medium at a concentration of 0.4% and 0.2% glucose as the carbon source. N.D.: not detected. The average values and standard deviations obtained from three separate experiments are represented. Comparisons are made as indicated after 2 h challenge: ** *p* < 0.01; * *p* < 0.05.

**Table 1 genes-11-00991-t001:** Bacterial strains and plasmids used in this study.

Strains or Plasmids	Genotype or Description	Source or Reference
Strains		
W3110	λ^−^ F^−^ derived from *E. coli* K-12	[34]
BW25113	*lacI*^q^*rrnB*_T14_ Δ*lacZ*_WJ16_*hsdR514* Δ*araBAD*_AH33_Δ*rhaBAD*_LD78_	[36]
SZ001	BW25113 *pgk*::Cm^r^	This study
SZ002	BW25113 *pykA*::Cm^r^	This study
SZ003	BW25113 *pykF*::Km^r^	This study
SZ004	BW25113 *pykA*:Cm^r^ *pykF*::Km^r^	This study
SZ005	BW25113 *gltA::*Cm^r^	This study
SZ006	BW25113 *pgi::*Cm^r^	This study
SZ007	BW25113 *pfo::*Cm^r^	This study
SZ008	BW25113 *glk::*Cm^r^	This study
WB001	W3110 *pgk*::Cm^r^	This study, W3110 × P1 (SZ001)
WB002	W3110 *pykA*::Cm^r^	This study, W3110 × P1 (SZ002)
WB003	W3110 *pykF*::Km^r^	This study, W3110 × P1 (SZ003)
WB004	W3110 *gltA*::Cm^r^	This study, W3110 × P1 (SZ004)
WB005	W3110 *pykA*::Cm^r^ *pykF*::Km^r^	This study, W3110 × P1 (SZ005)
WB006	W3110 *pgi*::Cm^r^	This study, W3110 × P1 (SZ006)
WB007	W3110 *pfo*::Cm^r^	This study, W3110 × P1 (SZ007)
WB008	W3110 *glk*::Cm^r^	This study, W3110 × P1 (SZ008)
Plasmids		
pKD3	*bla*, FRT, Cm^r^	[36]
pKD4	*bla*, FRT, Km^r^	[36]
pKD46	*bla*, *araC*, *gam-bet-exo*	[36]

^(^^1)^ Km^r^, resistant to kanamycin. Cm^r^, resistant to chloroamphenicol. ^(^^2)^ X P1, P1 transduction from BW25113.

**Table 2 genes-11-00991-t002:** Primers used in this study.

Primer Name	Sequence(5′→3′)
pykAP1	ATTTCATTCGGATTTCATGTTCAAGCAACACCTGGTTGTTTCAGTCAACGGAGTATTACTGTGTAGGCTGGAGCTGCTTCG
pykAP2	TGTTGAACTATCATTGAACTGTAGGCCGGATGTGGCGTTTTC GCCGCATCCGGCAACGTACCATATGAATATCCTCCTTAG
pykFP1	GCAGTGCGCCCAGAAAGCAAGTTTCTCCCATCCTTCTCAACTTAAAGACTAAGACTGTCTGTGTAGGCTGGAGCTGCTTCG
pykFP2	TTAAATAAAAAAAGCGCCCATCAGGGCGCTTCGATATACAAATTAATTCACAAAAGCAATACATATGAATATCCTCCTTAG
gltAP1	TGCGAAGGCAAATTTAAGTTCCGGCAGTCTTACGCAATAAG GCGCTAAGGAGACCTTAATGTGTAGGCTGGAGCTGCTTCG
gltA P2	AAAAATCAACCCGCCATATGAACGGCGGGTTAAAATATTTACAACTTAGCAATCAACCACATATGAATATCCTCCTTAG
pgk P1	TTTCAGGTAAGACGCAAGCAGCGTCTGCAAAACTTTTAGAATCAACGAGAGGATTCACCTGTGTAGGCTGGAGCTGCTTCG
pgk P2	AAAAATTGCGTGCTCTAAAAGCGCGCTGAAACAAGGGCAGGTTTCCCTGCCCTGTGATTTTCATATGAATATCCTCCTTAG
pgi p1	GTGACTGGCGCTACAATCTTCCAAAGTCACAATTCTCAAAATCAGAAGAGTATTGCTATGTGTAGGCTGGAGCTGCTTCG
pgi p2	TCAGGCATCGGTTGCCGGATGCGGCGTGAACGCCTTATCCGGCCTACATATCGACGATGACATATGAATATCCTCCTTAG
glk p1	ATGACAAAGTATGCATTAGTCGGTGATGTGGGCGGCACCAACGCACGTCTTGCTCTGTGTGTAGGCTGGAGCTGCTTCG
glk p2	TACAGAATGTGACCTAAGGTCTGGCGTAAATGTGCACCGGAACCGAGAAGGCCCGGCATATGAATATCCTCCTTAG
pfo p1	TTTCGTGCGCCCCTCATTTGCGCAATGTAAGGGTGTCATATGATTACTATTGACGGTAATTGTGTAGGCTGGAGCTGCTTCG
pfo p2	TTTAGAATTGGATAATCCTTATCCAGAGCATTTAATCGGTGTT GCTTTTCATATGAATATCCTCCTTAG
check for pgk KO	CCAATGAAGTCAACCTGTTGC
check for gltA KO	GATCCAGGTCACGATAACAAC
check for pykA KO	GAAGCAACGCTGGCAATTAC
check for pykF KO	CTGTAGCAATTGAGCGATG
check for pgi KO	CAGAGCGATACTTCGCTACTA
check for glk KO	CCGCAATACCCGTTAGCGTT
check for pfo KO	CTTCCTCTGATCTTCAAGCC
Km inbox	GTGAGATGACAGGAGATCCTG
Cm inbox	TCTTGCCCGCCTGATGAATGCTC
RT-glk F	CTGTATTGCCATCGCTTGCC
RT-glk R	TTACCTTCGACCGGTTCTGC
RT-pgi F	TTCATCGCTCCGGCTATCAC
RT-pgi R	CCAGGCTGAACGGAGTGATT
RT-pfkA F	TACATGGGTGCAATGCGTCT
RT-pfkA R	TAACGGCCCATCACTTCCAC
RT-fbaA F	ACTCCATCAACGCCGTACTG
RT-fbaA R	CAGTGGTCAGTGTGCAGGAT
RT-tpiA F	GCAAAACGTGGACCTGAACC
RT-tpiA R	ATGCACAGAACCGGAGTCAG
RT-gapA F	TTGACCTGACCGTTCGTCTG
RT-gapA R	GAAGTGCAAACTTCGCCGTT
RT-pgk F	CTCTCTGCTGCCGGTTGTTA
RT-pgk R	GCCTTTGTTGAAGCGAACGT
RT-gpmA F	ATCATCGCTGCACACGGTAA
RT-gpmA R	GCGATCTCGTCAGCATTACC
RT-eno F	TGGCGCGAAAACTGTGAAAG
RT-eno R	TGAACGCTTTGTTGCCTTCG
RT-pykA F	TCGCCTCTGACGTGGTAATG
RT-pykA F	GCATCACAGCGTCAGTACCA
RT-pykF F	GAACAGCCGTCTCGAGTTCA
RT-pykF R	TTAACAAGCTGCGGCACAAC
RT-pfo F	TTGCCACACATGCACTCTCT
RT-pfo R	CCTGCGGCATGAGATCAAGA
RT-gltA F	CGCATCCAATGGCAGTCATG
RT-gltA R	TTGCGCGGGTAAACAAATGG
RT-16S F	TACCGCATAACGTCGCAAGA
RT-16S R	AGTCTGGACCGTGTCTCAGT

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
