# Peer review of "Escherichia coli Increases its ATP Concentration in Weakly Acidic Environments Principally through the Glycolytic Pathway"

_genes, 2020, doi:10.3390/genes11090991_

Round 1
Reviewer 1 Report
In this manuscript, the authors examined which mechanisms of ATP production are used by Escherichia coli bacteria during adaptation to acidic conditions. It Is important as the E. coli survives in gastrointestinal tract due to its acid resistance mechanisms but not all of them are clearly investigated. Overall, the manuscript is well written, the introduction of the manuscript is sufficient but I do have some comments which can be found below.
Lines 35-37 The AR systems names should be mentioned
Lines 81-87 I do not understand why the conclusions of the manuscript are in the introduction. This paragraph should not be here and moreover, the aim of the study should be a bit elaborated.
The MM section is clear and the methods used are properly used but as the NGS-based analysis require high quality of the results I think that some Illumina run parameters and afterwards analysis should be included in this paragraph (i.e. RIN or equivalent values for RNA, run Q30, clustering, reads length, assembly parameters).
Figures 2 – 6. The figures descriptions are too long. In some of them, the MM fragments are repeated. I suggest to shorten the text under figures and move it to results paragraphs.
Figure 3. The A, B, C sections of the figure should be bigger because the GO terms are unreadable.
Figure 4. The scale on C and D section of the figure should be adjusted. The width of the columns also.
Figure 5. Same as for Figure 4, the scale of B and D sections should be the same.
Line 365-366 Please rephrase.
Lines 377 – 379 Should be “through” not “though” . Please rephrase “on its other consumed contents”.
I.e. Lines 381, 383, 395, 407 The E. coli species name should be written in italics. Please correct throughout whole manuscript.
The gene names should also be in italics, i.e. lines 399, 404, 419
Author Response
Point 1: Lines 35-37 The AR systems names should be mentioned.
Response 1: We have revised as suggested (lines 37-41). The four distinct AR mechanisms, the oxidative (OXI) system, glutamate decarboxylase (GAD) system, arginine decarboxylase (ARG) system, and lysine decarboxylase system are spelled out: “AR1 (oxidative system) requires the sigma factor RpoS and the cyclic AMP receptor protein CRP. AR2-AR4 functions through amino acid decarboxylation, AR2, AR3 and AR4 reliant on glutamate (glutamate decarboxylase system), arginine (arginine decarboxylase system) and lysine (lysine decarboxylase system), respectively.”
Point 2: Lines 81-87 I do not understand why the conclusions of the manuscript are in the introduction. This paragraph should not be here and moreover, the aim of the study should be a bit elaborated.
Response 2: We removed these sentences as suggested, and the aim of the study was elaborated on: "In this study, we attempt to clarify which pathway or genes are responsible for the increase in intracellular ATP concentration in E. coli grown in weakly acidic media with glucose as the sole carbon source. This will in turn reveal the mechanisms used by E. coli for adaptation to acidic environments. The expression of genes in the ATP synthesis pathway at different pH levels was analyzed, and the effects on AR and ATP concentration were investigated”.(Lines 84-88)
Point 3: The MM section is clear and the methods used are properly used but as the NGS-based analysis require high quality of the results I think that some Illumina run parameters and afterwards analysis should be included in this paragraph (i.e. RIN or equivalent values for RNA, run Q30, clustering, reads length, assembly parameters).
Response 3: The length is 30 bp for the pair end (PE) reads. If either end of a read was contaminated by more than 5bp of adapter sequence, both ends were removed. Low quality reads were defined as those with whose phred Quality value was less than or equal to 19, which accounted for more than 15% of all the Reads Bases. Both ends of the reads were filtered out when either of the PE reads was defined as low quality.
In addition, reads which contained more than 5% Ns were removed. Moreover, any reads that aligned to the rRNA sequence were also removed.
The clean reads were mapped to the reference genome by Rockhopper v2.03 using default parameters. Rockhopper’s approach aligned reads to the reference genome in the National Center for Biotechnology Information (NCBI) database, following that of Bowtie2. An FM-index was created for the genome based on the Burrows-Wheeler transform.
HTSeq v0.6.0 was run to calculate the counts of each gene, and gene expression levels were represented by the Reads per Kilobase per Million reads (RPKM) method, as described by Mortazavi et al [41].
We also added the following information to section 2.6 in the M&M under "RNA extraction and RNA-seq analysis”:
“The isolated RNA was sequenced on an Illumina HiSeq 2000 at Ribobio Co (China). Reads contaminated by adapter sequence were removed from the pair end (PE) reads (30 bp) when either of the PE reads was polluted. Low quality reads (phred quality≤19 in more than 15% of all the read bases) were further filtered out. Any reads that aligned to rRNA sequence were also filtered out. In addition, we removed reads that contained more than 5% Ns. The cleaned reads were aligned to the reference sequences in the National Center for Biotechnology Information (NCBI) database by Rockhopper v2.03 using default parameters. HTSeq v0.6.0 was utilized to calculate the co gen units of each e, and gene expression levels were represented by the Reads per Kilobase per Million reads (RPKM) method, as described by Mortazavi et al [41]. The RNA-seq analyses were repeated in two independent experiments using three biological replicates. Differentially expressed genes (DEGs) at either pH 5.5 or pH 7.5 were identified using the DEGseq method, using a cut off of fold change >2 and p value <0.01.”(Lines 161-172)
Point 4: Figures 2 – 6. The figures descriptions are too long. In some of them, the MM fragments are repeated. I suggest to shorten the text under figures and move it to results paragraphs.
Response 4: We have shortened the figure descriptions in figures 2-6 as suggested. The sentences which were repeated the M&M were deleted (Lines 225-235, 267-275, 304-310,332-337, 341-345).
Point 5: Figure 3. The A, B, C sections of the figure should be bigger because the GO terms are unreadable.
Response 5: We have made the figure more legible as suggested (figure 3).
Point 6: Figure 4. The scale on C and D section of the figure should be adjusted. The width of the columns also.
Response 6: We have redrawn these figures as suggested (figure 4).
Point 7: Figure 5. Same as for Figure 4, the scale of B and D sections should be the same.
Response 7: We have redrawn figure 5 as suggested (figure 5).
Point 8: Line 365-366 Please rephrase
Response 8: We have revised the paragraph to read: "We next examined whether these genes also affect acid resistance for cells in the stationary phase of growth. When the mutants were grown to stationary phase at pH 8 and then challenged for 2 hrs at pH 2.5, only ΔpykF showed significantly decreased survival (p<0.05) (Figure 5C). When the mutants were grown to stationary phase at pH 5.5 before challenging for 2 hrs at pH 2.5, both the pykF and pgk mutants showed lower survival. We noted that wild-type cells normally had a higher ATP content at pH 5.5 than that at pH 8 at the stationary phase. However, the intracellular ATP level of ΔpykF was significantly decreased in the stationary phase at both pH 8 and pH 5.5 (Figure 5D), while the pgk mutant showed a significantly decreased level only when grown to stationary phase at pH 5.5. These results confirmed that the glycolysis pathway mainly functions to supply ATP at a weakly acidic pH."(Lines 361-369)
Point 9: Lines 377 – 379 Should be “through” not “though” . Please rephrase “on its other consumed contents”.
Response 9: We have revised as suggested (line 379). The “on its other consumed contents” was rephrased to “on what other contents are being digested” (Lines 380-381).
Point 10: e. Lines 381, 383, 395, 407 The E. coli species name should be written in italics. Please correct throughout whole manuscript.
Response 10: We have corrected this throughout the manuscript as suggested (Lines 383,385, 397, 402,409, 412, 420, 424,429, 435, 438, 441, 451, and 455).
Point 11: The gene names should also be in italics, i.e. lines 399, 404, 419
Response 11: We have corrected this as suggested (Lines 386, 401, 406, 421, 422, 427, 432, 434, 435, and 464).

Reviewer 2 Report
In the manuscript "Escherichia coli increases its ATP concentration in weakly acidic environments principally through the glycolytic pathway", Zhang et al. build upon previous research on the relationship between intracellular ATP levels in E. coli and the environmental pH, convincingly showing that intracellular increase in ATP concentrations in weakly acidic environment originates in the glycolytic pathway.
My main concern is that the RNA sequencing results generated for this manuscript are not publically available, thus detracting from its reproducibility. Thus, to ensure the reproducibility of the results and increase the impact of the research, the authors are strongly encouraged to deposit generated RNA-seq short reads at the NCBI's Short Read Archive (https://www.ncbi.nlm.nih.gov/sra), as well as to deposit the calculated gene expression levels at the NCBI's Gene Expression Omnibus (https://www.ncbi.nlm.nih.gov/geo/).
Some of the minor remarks, for the author's consideration:
- Data shown in Figure 3A would be easier to interpret if visualised as the volcano plot (e.g. https://www.bioconductor.org/packages/release/bioc/vignettes/EnhancedVolcano/inst/doc/EnhancedVolcano.html). Such visualisation would also show the significance of the changes in gene expression.
- Figure 2A, 2C, 5B, 5D, 6B, and 6D: Is it essential to plot survival on the log-scale? In any case, the y-axis should never exceed survival of 100%.
- In the entire Discussion section, the italic formatting and the Greek letters (e.g. delta) are omitted (e.g. on the lines 419-420). Please revise.
- The comparison of the results with human cancer cells (line 465) is vague and ill-suited to being the last claim in the manuscript. Please expand on it and transfer it earlier in the Discussion, ending the manuscript more conclusively.
- Supplementary Figure 1 and 2 would be more informative if, in addition to colour-coding the genes, they would also show by how much the genes were up- or down-regulated.
- The description of the Supplementary Figure 1 does not describe what the figure shows (e.g. there is no x-axis in the figure). Should the description be more like that of Supplementary Figure 2?
- Line 359: weather >> whether
- Line 360-363: The sentence is unclear. Please reformulate.
- Line 373: greater death >> lower survival
Author Response
Point 1: Data shown in Figure 3A would be easier to interpret if visualised as the volcano plot (e.g. https://www.bioconductor.org/packages/release/bioc/vignettes/EnhancedVolcano/inst/doc/EnhancedVolcano.html). Such visualisation would also show the significance of the changes in gene expression.
Response 1: We have analyzed the data by volcano plot and revised figure 3 as suggested (Lines 267-270).
Point 2: Figure 2A, 2C, 5B, 5D, 6B, and 6D: Is it essential to plot survival on the log-scale? In any case, the y-axis should never exceed survival of 100%.
Response 2: As the survival of mutant and w3110 after challenge at pH 2.5 for 1-2 hours is from "None detected (<0.00001)" to near 100%, a log scale is needed to fit the data in one figure. We have redrawn figure 5 B, and changed the y-axis to not exceed 100%. Thank you for your suggestion.
Point 3: In the entire Discussion section, the italic formatting and the Greek letters (e.g. delta) are omitted (e.g. on the lines 419-420). Please revise.
Response 3: We have revised the italics formatting and the Greek letters as suggested.
Point 4: The comparison of the results with human cancer cells (line 465) is vague and ill-suited to being the last claim in the manuscript. Please expand on it and transfer it earlier in the Discussion, ending the manuscript more conclusively.
Response 4: We have deleted these sentences as they are just speculation and not important to this manuscript (Line 465).
Point 5: Supplementary Figure 1 and 2 would be more informative if, in addition to colour-coding the genes, they would also show by how much the genes were up- or down-regulated.
Response 5: We have added an explanation about the gene expression in supplementary figure 1 (Red genes are up-regulated more than 2 fold, blue genes show no significant change.) and figure 2 (Red genes are up-regulated more than 2 fold, blue genes show no significant change, and green genes are down-regulated more than 2 fold.). (Lines 480-481, 484-485)
Point 6: The description of the Supplementary Figure 1 does not describe what the figure shows (e.g. there is no x-axis in the figure). Should the description be more like that of Supplementary Figure 2?
Response 6: Yes, figure 1 should be described as figure 2. Thank you for your observation. We have revised the description in supplementary figure1 and supplementary figure 2 as suggested. Figure 1 was revised to say: “The up- and down-regulated genes in amino acid metabolic pathway. The changes in relative expression of E. coli amino acid metabolic genes (KEGG pathway) after reducing the pH level of the culture medium from 7.5 to 5.5 revealed the upregulation of genes belonging to particular AR systems.”(Lines 477-481). Supplementary figure 2 was revised to say: “Figure 2. The up- and down- regulated genes in the TCA cycle. Red genes are up-regulated, blue genes show no significant change, and green genes are down-regulated.” (Lines 483-485)
Point 7: Line 359: weather >> whether
Response 7: We have revised as suggested (Line 360)
Point 8: Line 360-363: The sentence is unclear. Please reformulate.
Response 8: We have revised the sentence as suggested: When the mutants were grown to stationary phase at pH 8 and then challenged for 2 hrs at pH 2.5, only ΔpykF showed significantly decreased survival (p<0.05) (Figure 5C). When the mutants were grown to stationary phase at pH 5.5 before challenging for 2 hrs at pH 2.5, both the pykF and pgk mutants showed lower survival”(Lines 361-364).
Point 9: Line 373: greater death >> lower survival
Response 9: We have revised as suggested (Line 375).

Round 2
Reviewer 1 Report
no other comments
Author Response
Thank you for your time and valuable suggestions.